# The Phosphorylation Status of Drp1-Ser637 by PKA in Mitochondrial Fission Modulates Mitophagy via PINK1/Parkin to Exert Multipolar Spindles Assembly during Mitosis

**DOI:** 10.3390/biom11030424

**Published:** 2021-03-13

**Authors:** Huey-Jiun Ko, Cheng-Yu Tsai, Shean-Jaw Chiou, Yun-Ling Lai, Chi-Huei Wang, Jiin-Tsuey Cheng, Tsung-Hsien Chuang, Chi-Ying F. Huang, Aij-Lie Kwan, Joon-Khim Loh, Yi-Ren Hong

**Affiliations:** 1Graduate Institute of Medicine, College of Medicine, Kaohsiung Medical University, Kaohsiung 807378, Taiwan; o870391@yahoo.com.tw (H.-J.K.); 4a1h0010@gmail.com (Y.-L.L.); aijliekwan@yahoo.com.tw (A.-L.K.); 2Department of Biochemistry, College of Medicine, Kaohsiung Medical University, Kaohsiung 807378, Taiwan; sheanjaw@kmu.edu.tw (S.-J.C.); cyhuang5@ym.edu.tw (C.-Y.F.H.); 3Ph.D. Program in Environmental and Occupational Medicine, College of Medicine, Kaohsiung Medical University and National Health Research Institutes, Kaohsiung 80708, Taiwan; moutzyy691010@yahoo.com.tw (C.-Y.T.); thchuang@nhri.edu.tw (T.-H.C.); 4Department of Neurosurgery, Kaohsiung Medical University Hospital, Kaohsiung 807378, Taiwan; 5Department of Biotechnology, Kaohsiung Medical University, Kaohsiung 807378, Taiwan; chwang@kmu.edu.tw; 6Department of Biological Sciences, National Sun Yat-sen University, Kaohsiung 80424, Taiwan; tusya@mail.nsysu.edu.tw; 7Immunology Research Center, National Health Research Institutes, Miaoli 35053, Taiwan; 8Department of Biotechnology and Laboratory Science in Medicine, Institute of Biopharmaceutical Sciences, National Yang-Ming University, Taipei 11221, Taiwan; 9Department of Medical Research, Kaohsiung Medical University Hospital, Kaohsiung 807378, Taiwan

**Keywords:** mitochondria, Drp1, PKA, phosphorylation, mitophagy, centrosomes, multipolar spindles

## Abstract

Mitochondrial fission and fusion cycles are integrated with cell cycle progression. Here we first re-visited how mitochondrial ETC inhibition disturbed mitosis progression, resulting in multipolar spindles formation in HeLa cells. Inhibitors of ETC complex I (rotenone, ROT) and complex III (antimycin A, AA) decreased the phosphorylation of Plk1 T210 and Aurora A T288 in the mitotic phase (M-phase), especially ROT, affecting the dynamic phosphorylation status of fission protein dynamin-related protein 1 (Drp1) and the Ser637/Ser616 ratio. We then tested whether specific Drp1 inhibitors, Mdivi-1 or Dynasore, affected the dynamic phosphorylation status of Drp1. Similar to the effects of ROT and AA, our results showed that Mdivi-1 but not Dynasore influenced the dynamic phosphorylation status of Ser637 and Ser616 in Drp1, which converged with mitotic kinases (Cdk1, Plk1, Aurora A) and centrosome-associated proteins to significantly accelerate mitotic defects. Moreover, our data also indicated that evoking mito-Drp1-Ser637 by protein kinase A (PKA) rather than Drp1-Ser616 by Cdk1/Cyclin B resulted in mitochondrial fission via the PINK1/Parkin pathway to promote more efficient mitophagy and simultaneously caused multipolar spindles. Collectively, this study is the first to uncover that mito-Drp1-Ser637 by PKA, but not Drp1-Ser616, drives mitophagy to exert multipolar spindles formation during M-phase.

## 1. Introduction

Mitosis, a process common to all eukaryotic organisms, is a set of highly orchestrated cellular events. Bipolarity of the mitotic spindles ensures the segregation essential for maintaining genomic stability. Defective chromosome segregation may result in chromosome instability (CIN), one of the major promoting factors of tumorigenesis resulting from centrosome overduplication or cytokinesis failure [1,2]. Mitotic catastrophe is a regulated antiproliferative process that occurs during defective or failed mitosis. Although it does not constitute a bona fide cell death mechanism in itself, mitotic catastrophe precedes antiproliferative measures including apoptosis, necrosis, and senescence to prevent the proliferation of defective mitotic cells [3]. Numeral and structural defects of the centrosomes prevalent in cancer contribute to tumorigenesis by promoting abnormal mitotic spindles and CIN or increasing tumor cell invasiveness [4,5]. The role of the centrosome is to form the bipolar mitotic spindles to split and separate the chromosome at G2/M phase [6,7,8]. Centrosome abnormalities result in the formation of multipolar spindles that promote chromosome segregation errors and genomic instability, contributing to aneuploidy and tumorigenesis [9,10].

In recent years, several studies have implicated mitochondria in centrosome functions, but communication between these organelles remains elusive. Mitochondria are double-membraned organelles that undergo fission and fusion cycles integrated with cell cycle progression, whose function is cellular energy production [11]. Mitochondrial electron transport chain (ETC) blockade has been shown to cause multipolar spindles leading to chromosome misalignment and chromosomal instability [6,12]. Defects in mitochondrial fission also lead to centrosome overduplication, misaligned chromosomes, and chromosome instability in metaphase [11,13]. The equilibrium of mitochondrial dynamics is regulated by fusion and fission. Mitochondrial fusion and fission are controlled by the associated dynamic proteins, Optic atrophy 1 (Opa1) and mitofusion 1/2 (Mfn1/2), as the fusion elements while dynamin-related protein-1 (Drp1) and fission 1 (Fis1) are involved in fission [14]. Loss of Drp1 cause mitochondrial hyperfusion and induced aneuploidy, revealing the importance of mitochondrial dynamics in maintaining cell division [11]. Fused mitochondria become more fragmented by increased phosphorylation of Drp1-Ser616 through Cdk1/Cyclin B, required for inheritance through the G2/M phase of the cell cycle [15]. Recent studies have shown that mitophagy is a specific selective type of autophagy that promotes mitochondrial turnover and prevents the accumulation or maintains the cellular homeostasis of dysfunctional mitochondria [11,16]. In addition, knockdown of Drp1-induced mitochondrial hyperfusion triggers replication stress and defects in chromosome segregation during mitosis. They also observed that defects in mitochondrial fission lead to centrosome overduplication, misaligned chromosomes, and chromosome instability in metaphase [11]. Moreover, blockade of ETC complex I causing overexpression of Aurora A, Polo-like kinase 4 (Plk4), and Cyclin E has been observed with the emergence of amplified centrosomes [6]. Treatment with rotenone (ETC complex I inhibitor) in HeLa cells blocks mitosis and inhibits cell proliferation by suppressing the reassembly of microtubules [17]. A recent review also reported that mitochondrial proteins can influence cell cycle progression, alter chromosome morphology, and localize to centrosomes based on some previous studies [18]. On the other hand, Aurora A and Plk1 locate and phosphorylate LARA and Miro in mitochondria, respectively [19,20,21]. Although these studies have implicated mitochondria in centrosome function, the crosstalk between mitochondria and centrosome organelles remains elusive. For example, how does the joining of mitochondrial ETC proteins and centrosomal proteins play a role in bipolar spindles formation? How do mitochondria ETC inhibitors and Mdivi1, a specific Drp1 inhibitor, affect mitochondrial dynamics involving mitotic kinases/proteins and drive mitophagy to exert multipolar spindles? How does mitophagy drive centrosome dysfunction leading to multipolar spindles, and are the centrosome aberrations the cause or consequence of mitophagy?

In this study, we demonstrated that ETC inhibitors or Mdivi1 drive mitophagy to induce multipolar spindles formation. We also found that the phosphorylation status of Ser637, but not Ser616, in Drp1via PKA converges with regulatory mitotic kinases Cdk1, Plk1, Aurora A, and associated proteins like Cyclin B, Bora, AIBp, CEP192/CEP215, and TACC3/chTOG to induce multipolar spindles formation in the mitotic phase (M-phase). This is the first study to reveal that the status of Drp1 phosphorylation at Ser637 in mitochondria by protein kinase A (PKA) (herein designated mito-Drp1-Ser637) plays a vital role in mitochondrial fission via the PINK1/Parkin pathway to drive mitophagy and, via mitotic kinases, induces multipolar spindles formation in the M phase.

## 2. Materials and Methods

### 2.1. Cell Culture and Cell Cycle Synchronization

HeLa cells were cultured in Dulbecco’s modified Eagle’s medium (DMEM, Thermo Fisher Scientific, Inc., Waltham, MA, USA) supplemented with 10% FBS, 2 mM L-glutamine and 1% penicillin/streptomycin (DMEM, Thermo Fisher Scientific, Inc., Waltham, MA, USA). Cells were maintained at 37 °C in a humidified atmosphere of 5% CO_2_. Concentrations and time courses of different G2/M synchronization and release times are presented in Appendix A; for gene targeting and other experiments, cells underwent 16 h incubation with nocodazole (Sigma-Aldrich; MO, USA) at 200 ng/mL followed by 1 h release, and were then resuspended in prewarmed media at 37 °C containing ETC inhibitors ROT and AA, Mdivi-1 or Dynasore for 24 h.

### 2.2. In Vitro Cytotoxicity Assay

Briefly, cells were cultured in 96-well plates and stimulated with different concentrations of ROT, AA, Mdivi-1or Dynasore for 24, 48, or 72 h. The cell viability of each treated sample was measured using the CCK-8 kit (CCK-8, Sigma 96992, St. Louis, MO, USA) according to the manufacturer’s instructions. The absorbency of cells was measured using a 96-well plate reader at 450 nm.

### 2.3. Oxygen Consumption Rate (OCR)

The oxygen consumption rate (OCR) of cells (1×10^6^ cells/mL in medium) was measured by high-resolution respirometry (Oroboros O_2_k, Innsbruck, Austria). Mitochondrial respiration was measured on the basis of the decrease in O_2_ consumption. The total O_2_ consumption of the cells measured at baseline represented “routine.” After the addition of oligomycin (Oma; 2 μM), the remaining rate of mitochondrial respiration represents a proton “leak.” Then the maximum “ETS” (electron transfer system) was measured after the addition of the uncoupling reagent FCCP (1 μM) to induce maximal O_2_ consumption. The difference between the basal and maximal respiration is called the reserve capacity of OCR. Non-mitochondrial respiration, the “ROX” (residual oxygen consumption), was quantified by inhibiting mitochondrial respiration by the addition of ROT (2 μM) and AA (2 μM).

### 2.4. Measurement of Mitochondrial Membrane Potential (ΔΨm)

Tetramethylrhodamine methyl ester (TMRM) is a monochromatic probe or single wavelength dye that is readily sequestered by polarized/energized mitochondria. Cells were incubated at room temperature for 40 min at 37 °C with 25 nM TMRM (I34361, Invitrogen Life Technologies, Carlsbad, CA, USA) and analyzed by flow cytometry and fluorescence live cell microscopy. Images were acquired using a Zeiss 700 VIS CLMS confocal microscope equipped with a ×40 oil immersion objective (Zeiss, Oberkochen, Germany).

### 2.5. ATP Measurement

Total cellular ATP levels were measured using a bioluminescence assay (Life Technologies, CA, USA). Cell lysates were added to a reaction buffer containing 25 mM Tricine (pH 7.8), 0.5 mM d-luciferin, 1.25 µg/mL firefly luciferase, 5 mM MgSO_4_, 100 µM EDTA, and 1 mM DTT, then incubated in the dark for 15 min at room temperature. Luminescence was measured using a luminometer at 560 nm. A standard curve was generated and used to calculate ATP concentration.

### 2.6. Flow Cytometry

Cytoplasmic hydrogen peroxide and mitochondrial superoxide levels and ΔΨm were measured by staining with H_2_-DCFDA (5 µM), MitoSOX red (5 µM), and JC-1 (5 µM), respectively, in HBSS for 30 min at 37 °C under 5% CO_2_. Cells were then washed with HBSS, resuspended in ice-cold phosphate-buffered saline (PBS) and analyzed by flow cytometry using a fluorescein isothiocyanate filter (530 nm) for DCFDA, a phycoerythrin filter (575 nm) for MitoSOX red and both fluorescein isothiocyanate and phycoerythrin filters for JC-1 detection. Cells treated with 0.3% H_2_O_2_ or 500 µM doxorubicin were used as positive controls for the detection of cytoplasmic hydrogen peroxide and mitochondrial superoxide, respectively.

### 2.7. Cell Cycle Analysis

The cells were harvested and re-suspended in 70% ethanol in phosphate buffered saline (PBS) and incubated at −20 °C for 1 h. Subsequently, cells were re-suspended in cold PBS containing 20 μg/mL PI and 100 μg/mL RNAse A (Sigma-Aldrich, St. Louis, MO, USA). The cells were then incubated in a dark chamber for 30 min at room temperature, after which the DNA content was analyzed by BD FACScan using FACS Diva software (BD Biosciences, San Jose, CA, USA).

### 2.8. Immunofluorescence Staining and Confocal Microscopy

To observe the changes in mitochondrial morphology and mitotic spindles, HeLa cells were treated with mitochondrial ETC complex inhibitors for 24 h to observe the G2/M phase of mitochondria or mitotic spindles formation, then synchronized with nocodazole for 24 h and released with DMEM into M phase. The mitochondria were stained with 100 nM MitoTracker Red for 30 min and then washed with PBS, fixed in 3.7% paraformaldehyde (PFA) for 15 min and permeabilized with 0.1% Triton for 15 min at room temperature. DNA was stained with 4, 6-diamidino-2- phenylindole (DAPI, 2 mg/mL). Mitotic spindles morphology was detected by probing with mouse anti-α-tubulin antibody (1:1000). The secondary antibodies were rhodamine-conjugated goat anti-mouse antibodies (1:5000). DNA was stained with 4, 6-diamidino-2- phenylindole (DAPI, 2 mg/mL). Immunofluorescent imaging was performed using a FluoView 1000 confocal laser scanning microscope (Olympus, Tokyo, Japan). To quantify the structural mitochondrial network fragmentation, the images of mitochondria were started with background substrate, filtered (median), thresholded, and binarized to identify the mitochondrial segments using Image J. Then the number of fragmentation mitochondria was calculated and divided by the total pixel of mitochondria (area) to yield the mitochondrial fragmentation index.

### 2.9. Mitochondria Subcellular Fractionation and Preparation

Cells were cultured in adherence as described above. To fractionate cells into cytosol and mitochondrial fractions, a subcellular protein fractionation kit was used following the manufacturer’s protocol. The separation of mitochondrial outer membrane and matrix was performed following the protocol steps. After isolation, cytosol and mitochondrial pellets were resuspended in SDS-PAGE buffer and analyzed by Western Blot.

### 2.10. Western Blot Analysis

For Western blot analysis, HeLa cells were maintained in DMEM supplemented with 10% FBS. Cells were harvested 24 h after ROT and AA treatment. After that, the cells were re-suspended in cell lysate buffer [50 mM Tris-HCl (pH 7.8), 150 mM NaCl, 5 mM EDTA, 0.5% Triton X-100, 0.5% Nonidet P-40, 0.1% deoxycholate, and leupeptin, aprotinin, and 4-(2-aminoethyl) benzenesulfonylfluoride (10 mg/mL each)]. Samples were sonicated and then centrifuged at 14,000 rpm for 15 min at 4 °C. The supernatant was then placed into a fresh centrifuge tube, protein sample buffer was added, and the sample was heated to 95 °C for 5 min. Proteins were separated by 10–12% SDS/PAGE, transferred to Hybond PVDF membranes (Amersham, GE Healthcare, Velizy-Villacoublay, France), blocked with 5% milk in PBS with 0.05% Tween 20 (PBS/Tween) for 1 h, washed with PBS/Tween 20 three times for 10 min each, and incubated with the relevant antibodies. The blots were washed with PBS/Tween 20 three times for 10 min each, and then HRP-conjugated antibody was used as the secondary antibody for an additional 1 h.

### 2.11. Statistical Analysis

Data are presented as mean ± standard deviation. Statistical analyses were performed using one-way analysis of variance. Data were compared using Student’s *t*-test. The level of statistical significance was set at * *p* < 0.05, ** *p* < 0.01, *** *p* < 0.001.
**Reagent Type (Species) or Resource****Designation****Source or Reference****Identifiers****Additional Information**Cell line(H. sapiens)HeLaATCC (Manassas, VA, USA)CCL-2Maintained in DMEM supplemented with 10% FBS, 1% penicillin/streptomycin and 1% L-GlutamineAntibodyRabbit anti-Aurora AAbnova (Taipei, Taiwan)PAB03591:1000AntibodyRabbit anti-Aurora-A(T288)Abnova (Taipei, Taiwan)MAB72991:1000AntibodyRabbit anti-AIBpAbcam (Cambridge, MA, USA)ab1220141:500AntibodyRabbit anti-Plk1 Cell signaling (Beverly, MA, USA)s45131:1000AntibodyRabbit anti-Plk1(Thr210)Cell signaling (Beverly, MA, USA)S54721:1000AntibodyMouse anti-Drp1BD (BD Biosciences, San Jose, CA, USA)6111121:1000AntibodyMouse anti-Drp1 (Ser 616) Cell signaling (Beverly, MA, USA)S34551:1000AntibodyMouse anti-Drp1 (Ser 637)Cell signaling (Beverly, MA, USA)S48671:1000AntibodyRabbit anti-Opa1Cell signaling (Beverly, MA, USA)S675891:1000AntibodyRabbit anti-Mfn1Cell signaling (Beverly, MA, USA)S147391:1000AntibodyRabbit anti-Mfn2Cell signaling (Beverly, MA, USA)S94821:1000AntibodyRabbit anti-cdc (Cdk1)Cell signaling (Beverly, MA, USA)S284391:1000AntibodyRabbit anti-cdc2 (Thr14)Cell signaling (Beverly, MA, USA)S25431:1000AntibodyRabbit anti-Cyclin BCell signaling (Beverly, MA, USA)S122311:1000AntibodyRabbit anti-Cyclin B (S133)Cell signaling (Beverly, MA, USA)S41331:1000AntibodyRabbit anti-VDAC1Invitrogen (Life Technologies, CA, USA)PA1-954A1:1000AntibodyRabbit anti-ch-TOG (CKAP5)GeneTex (Irvine, CA, USA)GTX306931:1000AntibodyRabbit anti-TACC3Santa Cruz (Santa Cruz, CA, USA)SC-227731:500AntibodyRabbit anti-TACC3 (Ser 558)Cell signaling (Beverly, MA, USA)S88421:1000AntibodyMouse anti-P62 Abcam (Cambridge, MA, USA)Ab564161:2000AntibodyRabbit anti-LC3BCell signaling (Beverly, MA, USA)S38681:1000AntibodyMouse anti-GAPDHArigo (Hsinchu, Taiwan)ARG101121:5000AntibodyRabbit anti-PINK1Elabscience (Houston, TX, USA)E-AB-173711:1000AntibodyRabbit anti-PRKNElabscience (Houston, TX, USA)E-AB-634901:1000AntibodyMouse anti-α-tubulinSigma-Aldrich (St. Louis, MO, USA)T51681:5000AntibodyRabbit anti-γ-tubulinSigma-Aldrich (St. Louis, MO, USA)T51921:5000AntibodyTotal OXPHOS rodent WB antibody cocktail (Mouse)Abcam (Cambridge, MA, USA)ab1220141:1000Commercialassay or kitMitochondria Isolation KitThermo Fisher (Waltham, MA, USA)89874
Commercialassay or kitTMRM Reagent Invitrogen (Life Technologies, CA, USA)I34361Mitochondrial membrane potential indicatorCommercialassay or kitCellROX™ Green ReagentInvitrogen (Life Technologies, CA, USA)C10444Oxidative stress detectionCommercialassay or kitATP Assay KitSigma-Aldrich (St. Louis, MO, USA)MAK190ATP Colorimetric/Fluorometric Commercialassay or kitCell counting Kit-8Sigma-Aldrich (St. Louis, MO, USA)96992Proliferation and cytotoxicity assaysCommercialassay or kitMitoTrackerInvitrogen (Life Technologies, CA, USA)M7512


## 3. Results

### 3.1. Mitotic Spindles Multipolarity Induced by ROT and AA Shows Loss of Spindles Pole Integrity

Centrosomal numerical abnormalities are recurrent features of human tumors. Several studies have shown that blockade of ETC causes centrosome amplification leading to chromosome misalignment and CIN [6,17]. Here, we re-visited how mitochondrial ETC inhibition disturbed mitosis progression in HeLa cells. We first investigated the mitotic effects induced in HeLa cells after treatment with nocodazole. Analyses of nuclear morphology showed that nocodazole-treated cells exhibited chromosomal condensation and segregation, characteristic morphological changes seen in cells blocked in mitotic prometaphase (Appendix A). Flow cytometry analysis showed that nocodazole (200 nM) induced characteristic G_2_/M-pattern cell cycle arrest (Appendix A). We next conducted a detailed analysis of the levels of two cell cycle proteins, Cyclin B and Cdk1. Their levels started to increase at 16 h after nocodazole treatment (Appendix A). A growth curve was generated in the presence of rotenone (ROT) or antimycin A (AA) by using cell counting Kit-8 (CCK-8) analysis for the cell growth rate. In comparison to control cells, ETC inhibitor-treated HeLa cells revealed a quite different growth pattern, growing slowly at 24 h after treatment in a dose-dependent manner. The half maximal inhibitory concentration (IC_50_) values of ROT and AA in the HeLa cells were 0.14 ± 0.02 and 11.9 ± 0.5 μM at 24 h, respectively (Appendix A). Thus, all subsequent experiments were performed using 200 nM ROT or 10 μM AA. Additionally, 72 h of treatment with ROT or AA seriously affected the respiration energy necessary for cell growth.

To analyze the respiratory function, cellular OCR was measured using an Oxygraph-2k oxygen electrode. A suspension of cells (1x10^6^ cells/mL) in culture medium was added to the Oxygraph-2k chambers. OCR studies showed that HeLa cells lost almost all respiratory capacity after ROT or AA treatments (Figure 1A). The PCP protocol (phosphorylation control protocol) was applied to evaluate the cellular routine respiratory state, the proton leak after inhibition of ATP synthesis, the mitochondrial maximum electron transfer system (ETS) coupling state, and the ROX in ROT- and AA-inhibited respiration after sequential inhibition of complexes I and III. The difference in OCR between the control and inhibitor-treated cells was used to evaluate the relationship between respiration and mitotic spindles aberrance.

The normal mitochondrial functions are essential for centrosomal homeostasis and spindles formation in G2/M phase, including dynamic mitochondrial oxidative phosphorylation (OX-PHOS) and ATP production. To further characterize the relationship between mitochondria and centrosomal function, we first examined the mitotic spindles formation after blocking ETC complexes I and III by ROT or AA respectively for 24 h. We microscopically visualized the inhibitor-treated cells and found that blockade of ETC complexes I and III led to aberrant mitotic performance. The amplified centrosomes increased tripolar spindles formation by 10% and 11% and multipolar spindles formation by 21% and 20% respectively in the presence of ROT and AA in comparison to the control cells (Figure 1B). 

Aneuploidy-associated stressors (proteotoxic, metabolic, oxidative stress, and altered levels of replication factors) due to gene dosage imbalance will, in turn, fuel replication stress, thus completing the CIN propagation circle. In addition, aneuploidy-associated stressors such as proteotoxic or oxidative stress, which scale with aneuploidy, can exacerbate replication stress to further induce mitotic aberrations and DNA damage [22]. Recent research also shows that exogenous or endogenous reactive oxygen species (ROS) promoted mitotic arrest. Delayed formation and abnormal function of the mitotic spindles directly impeded mitosis and promoted abnormal chromosome separation, and was responsible for ROS-induced mitotic arrest [23]. We next examined oxidative stress by flow cytometry in cells challenged with ROT and AA. ROS levels were measured using two different methods, CellROX and dihydroethidium (DHE) assays. Addition of ROT and AA led to a significantly higher production of ROS compared to untreated controls in both assays (Appendix A). Since it is well established that impairment of mitochondrial respiratory chain complexes, in particular complexes I and III, induces superoxide anion formation, we determined whether ROT or AA affected mitochondrial function. We estimated ΔΨm by quantifying the mitochondrial fluorescence intensity of TMRM. We observed a small decrease in mitochondrial TMRM fluorescence after ROT or AA. However, the difference was not significant (Appendix A). In addition, the cellular ATP level was decreased by ROT or AA, reaching 50% lower than that of the untreated cells (Figure 1C).

To investigate the mitochondrial morphological changes from ETC blockade, we analyzed the mitochondrial dynamic associated proteins Drp1, Mfn1, Mfn2, and Opa1. We also used confocal microscopy to evaluate the mitochondrial morphological changes by staining with MitoTracker Red. Image analysis suggests that mitochondria undergo more fragmentation in ROT or AA-treated HeLa cells compared to Mock (Figure 1D). Thus, whether ROT or AA induces multipolar spindles formation is relevant to dysfunctional mitochondrial fission. As shown in Figure 1E, ROT or AA decreased Drp1 expression, but increased the Drp1 Ser637/Ser616 phosphorylation ratio, and increased Mfn1expression, but did not affect Opa1 and Mfn2 expression. Altogether these data demonstrate that 24 h of treatment with ROT or AA significantly impairs mitochondrial respiratory chain function, with a decline in ATP production accompanied by increasing oxidative stress in HeLa cells.

### 3.2. Mdivi-1 Induces Multipolar Spindles Formation by Increased Oxidative Stress due to Mitochondrial Fragmentation and Dysfunction

It was evident that defects in mitochondrial fission led to defects in chromosome segregation during mitosis [11]. We also noted that ETC inhibitors might have pleiotropic effects. Next, we examined the effect of Mdivi-1 treatment on multipolar spindles formation and mitochondrial morphology in mitotically arrested cells. First, the effect of Mdivi-1 on the growth of HeLa cells was examined by CCK-8 assay. As shown in Appendix A, Mdivi-1 decreased HeLa cell growth in a dose-dependent manner and markedly reduced HeLa cell viability in a time-dependent manner compared to untreated cells, with IC_50_ values of 48.3 ± 2.3 μM, 36.24 ± 1.4 μM, and 22.27 ± 3.6 μM for 24, 48, and 72 h respectively.

Examining microtubule and chromosome morphology, we found that in Mock cells microtubules re-polymerized and the cells underwent bipolar division within 16 h post release (Figure 2A). Dose-dependent mitotic spindles multipolarity and misaligned chromosomes were also observed in mdivi-1-treated HeLa cells that were synchronized by nocodazole pretreatment (Figure 2A). To determine whether Mdivi-1 treatment induced ROS production in cultured HeLa cells, ROS levels were measured using DHE staining and cellROX. Following 24 h incubation with nocodazole-induced mitotic arrest, different concentrations of Mdivi-1 (20 and 50 μM) were added, and ROS production was determined by immunofluorescence and flow cytometry (Appendix A). The microimages in Appendix A demonstrate that oxidative stress was markedly increased in a dose-dependent manner in Mdivi-1treated cells compared with cells in nocodazole-induced mitotic arrest, thus confirming the effects of Mdivi-1 on nocodazole-induced multipolar spindles via decreased ATP levels and ΔΨm (Figure 2B and Appendix A).

Although Mdivi-1 reduces mitochondrial fission, increases mitochondrial fusion, and maintains mitochondrial function [24], we observed an increase in mitochondrial fragmentation in Mdivi-1 treated cells in nocodazole-induced mitotic arrest (Figure 2C). To determine the effect of Mdivi-1 on Drp1 and phosphorylated Drp1 at Drp1-Ser616 and Drp1-Ser637, Opa1, Mfn1, and Mfn2, we treated cells in nocodazole-induced mitotic arrest for 24 h with two different concentrations of Mdivi-1 (20 and 50 μM). As shown in Figure 2D, we found significantly reduced Drp1 and phosphorylated Drp1-Ser616, and increased phosphorylated Drp1-Ser637, Mfn2, and Opa1. These results indicate that the G2/M cell cycle arrest and aneuploidy observed with Mdivi-1 cannot be attributed to change in total ATP production or mitochondrially generated ROS. Moreover, Mdivi-1 induces multipolar spindles by increased oxidative stress, mitochondrial fragmentation, and dysfunction.

### 3.3. Dynasore Has a Slight Effect on Mitotic Defects, but Does Not Affect Mitochondrial Morphology

Dynasore, a cell-permeable small molecule that inhibits dynamin activity and non-competitively inhibits the GTPase activity of dynamin, has been widely studied in endocytosis and phagocytosis [25]. Human HeLa cells were treated with increasing concentrations of Dynasore, and then the cell viability was assessed by CCK-8 kit at 24, 48, and 72 h as shown in Appendix A. To evaluate whether Dynasore affected multipolar spindles formation and mitochondrial function, cells were exposed to Dynasore (40 and 80 μM) for 24 h. The effects of Dynasore were analyzed by observing multipolar spindles. Dynasore slightly promoted multipolar spindles formation in the M phase (Figure 3A).

As shown in Appendix A, DHE assay revealed that high-dose Dynasore (80 μM) significantly increased superoxide anion (O_2_^•−^) fluorescence, but reduced cellROX fluorescence by 19% when compared with M phase alone. Mitochondrial functions such as TMRM fluorescence and ATP production were also measured after treatment with Dynasore. As exhibited in Appendix A and Figure 3B, we saw no effect of Dynasore on ATP production and ΔΨm fluorescence compared with Mock (Figure 3C). Dynasore did not affect mitochondrial morphology in M phase (Figure 3C), with the exception of an increase in phosphorylated Drp1-Ser637, reduced phosphorylated Drp1-Ser616, and increased Mfn 1 and Opa1 in the presence of 80 μM Dynasore (Figure 3D). These results indicate that Dynasore did not significantly affect mitotic defects and mitochondrial dysfunction.

### 3.4. Multipolar Spindles Induction in Mitotic Cells Is Associated with Mitochondrial ETC Blockade

The oxidative phosphorylation system (OXPHOS) is essential for cellular energy metabolism. To clarify crosstalk between centrosome and mitochondria in HeLa cells, and differences in OXPHOS protein levels in HeLa cells following ROT, AA, Mdivi-1, and Dynasore treatment, we used Western blot analysis to examine the mitochondrial OXPHOS complexes because they directly affect the mitochondrial function. At basal conditions, OXPHOS protein levels were lower following ROT compared to control cells. Significantly decreased levels of complex I proteins were found as compared to control cells. AA treatment induced significantly decreased expression of complex I, complex III, and complex IV. Interestingly, Mdivi-1 significantly reduced levels of complex I, complex III, and complex IV proteins, but there were no appreciable dynasore-induced changes in protein levels (Figure 4A). 

Mitochondrial function was assessed in ROT, AA, Mdivi-1, and Dynasore treated and untreated cells by measuring oxidative stress, ΔΨm and ATP. As Figure 4B shows, significantly reduced levels of ATP production were found in HeLa cells treated with ROT, AA, and Mdivi-1 relative to untreated HeLa cells. In contrast, ATP production was unchanged in Dynasore-treated HeLa cells. ΔΨm was a bit different from ATP production. While ROT, AA, Mdivi-1, and Dynasore significantly increased ROS overproduction in HeLa cells, ROT, AA, and Mdivi-1 increased both ROS and superoxide in HeLa cells, while Dynasore only slightly increased ROS (Figure 4C,D). ΔΨm fluorescence was significantly increased in HeLa cells treated with ROT, AA, and Mdivi-1, but Dynasore-treated cells showed no change in ΔΨm fluorescence (Figure 4E). In this study, through detailed characterization of the effect of ROT, AA, Mdivi-1, and Dynasore on cell cycle progression, we found that ROT, AA and Mdivi-1 induced a time-dependent accumulation of G2/M phase cells in the HeLa strain. Importantly, under treatment conditions that cause significant M-phase arrest in HeLa cells (ROT, AA and Mdivi-1 for 16 to 24 h), Dynasore did not induce M-phase arrest in HeLa cells. In control cells nocodazole (200 nM) induced a characteristic G2/M-pattern cell cycle and also did not show M phase accumulation after extended exposure for up to 24 h (Figure 4F). These results suggest that ROT, AA, and Mdivi-1 induce M-phase arrest in a tumor-specific manner and multipolar spindles formation is affected by mitochondrial complexes I, III, and IV.

### 3.5. PKA Plays a Key Role to Switch the Phosphorylation Status of Mito-Drp1-Ser637 Rather than Mito-Drp1Ser616 and Triggers Multipolar Spindles Formation in Mitochondria during Mitotic Arrest

It is well-known that Drp1-Ser637 is phosphorylated by PKA [26]. We therefore performed the same experiments with mitochondrial inhibitors along with forskolin (FSK, PKA activator) or H89 (PKA inhibitor) to examine the biological effects. Our data showed that ROT, AA, and Mdivi-1 with FSK increased mito-Drp1-Ser637 levels in both interphase and M-phase (Figure 5A), whereas H89 prevented Drp1-Ser637 as expected (Figure 5C). We noted that mito-Drp1-Ser637 was the dominant form rather than normal Drp-Ser616 in M-phase (Figure 5A, M-phase panel, lanes 4, 5, 6). It should be also noted that ROT, AA, and Mdivi-1 all increased mitochondrial mito-Drp1-Ser637 levels at mitosis (Figure 5C, lanes 2, 3, 4 in mitochondria panel). Moreover, the confocal microscopy images were also consistent with the expected data using Dynasore with FSK to trigger multipolar spindles formation (Figure 5B) and Mdivi-1 with H89 to reduce multipolar spindles formation (Figure 5D). Altogether, these results further affirm that PKA truly was involved in mitochondrial dysfunction and caused mitotic defects during mitosis. We thus believe that Drp1-Ser616 fission activity regulated by Cdk1/Cyclin B is a prerequisite for normal equal mitochondria partition and faithful distribution into two dividing cells at M-phase (Figure 5A,B). However, when cells were insulted by ROT, AA, and Mdivi-1 inhibitors, the phosphorylation status evoked mito-Drp1-Ser637 by PKA rather than mito-Drp1-Ser616 by Cdk1/Cyclin B in mitochondrial fission (Figure 5C,D).

### 3.6. The Phosphorylation Status of Ser637, but Not Ser616, Modulates Mitophagy via the PINK/Parkin Pathway to Induce Multipolar Spindles Formation

Drp1 normally resides in cellular cytosol and it is the mitochondrial Drp1 that participates in mitochondrial fission. We initially investigated whether mitochondrial ETC complex inhibitors (ROT, AA and Mdivi-1) decreased Drp1 expression and the Drp1-Ser637/Ser616 phosphorylation ratio in mitotic arrest, clarifying the role of mitochondrial ETC signaling in Drp1 activation. Given that Drp1 translocation to mitochondria can be induced by Drp1 phosphorylation [27], we then conducted subcellular fractionation experiments and found that nocodazole treatment induced Drp1 phosphorylation at Ser616 and caused its subsequent recruitment to the mitochondria (Figure 6A). Conversely, mito-Drp1-Ser637 was decreased in the mitochondrial fraction in nocodazole-treated cells relative to untreated cells. In addition, we also found decreased Drp1-Ser616 and total Drp1 and increased mito-Drp1-Ser637 levels in mitochondria after ROT, AA, and Mdivi-1 in combined nocodazole-treated cells (Figure 6A,B). To determine whether Drp1 translocation by ETC complex inhibitors stimulates mitochondrial fragmentation by promoting mito-Drp1-Ser637 phosphorylation, cells were exposed to Dynasore (40 and 80 μM) for 24 h. There was a significant stimulation of mitochondria Drp1 Ser616 phosphorylation and mitochondria total Drp1 by Dynasore treatment relative to mock cells, and significantly increased mito-Drp1-Ser637 (Figure 6C). Thus, our results suggest that mitochondrial ETC signaling induces the phosphorylation of mito-Drp1-Ser637 and causes its subsequent translocation to mitochondria, offering an alternative mechanism for ROT, AA, and Mdivi-1 induced mitochondrial fragmentation.

Impaired mitochondrial function leads to the production of excessive ROS, which induces oxidative stress and damages the mitochondrial respiratory chain. Mitophagy targets damaged mitochondria for selective elimination, a process triggered by the PINK1/Parkin signaling cascade [28]. In the present study, we found significantly higher expression of PINK1 and Parkin and protein abundance of P62 and LC3B-II in the M phase arrest induced by ROT, AA, and Mdivi-1 treatment compared to Mock (Figure 6D). Conversely, the protein abundance of mitophagy initiators PINK1, Parkin, P62, and LC3B-II was lower in the M phase arrest induced by Dynasore (Figure 6D). The results showed that ROT, AA, and Mdivi-1 treatment can upregulate the expression of Parkin and promote mitophagy in M phase arrest. Next, we sought to determine the effect of ROT, AA, and Mdivi-1 on PINK1, Parkin, P62, and LC3B-II subcellular distribution. Because both ETC blockade and an increase in PINK1 expression promote Parkin translocation, we wondered if mitochondrial ETC blockade could enhance PINK1 expression. However, because the protonophoric uncoupler carbonyl cyanide-chlorophenyl hydrazine (CCCP) triggers PINK1/Parkin translocation, we reasoned that any effect a blockade of mitochondrial ETC might have on PINK1/Parkin to be post-translated in nature. Consistent with this view, we found that untreated HeLa cells were the only cells of the varied cell types used in this work in which endogenous mitochondrial PINK1 was detectable, though an endogenous PINK1/Parkin pathway was seen clearly with higher exposure to 10 μM CCCP. ROT, AA, and Mdivi-1 increased mito-PINK1, mito-Parkin, mito-P62, and mito-LC3B-II expression (Figure 6E). Remarkably, although Dynasore does not affect PINK1/Parkin-mediated mitophagy, Dynasore caused strongly enhanced autophagy in non-treated HeLa cells (Appendix A). The results of this study were consistent with previous studies showing that 10 μM CCCP decreased mito-Drp1-Ser616 and mito-Drp1 and increased mito-Drp1-Ser637 levels at mitosis (Figure 6F), further confirming the involvement of PINK1/Parkin in mito-Drp1-Ser637-induced mitophagy. 

### 3.7. Mitotic Kinases Regulating Centrosome Maturation and Mitotic Spindles Assembly Are Associated with Mitochondrial ETC Blockade

Cdk1 activity is regulated by removal of inhibitory phosphorylation of Cdk1 in addition to increased Cyclin B expression. To examine Cdk1 and Cyclin B during the G2/M transition, we performed immunoblotting after synchronization at the G2/M border. When the cells were released from the G2/M border, Cyclin B levels in control cells gradually increased, which is indicative of normal cell cycle progression (Figure 7A). To analyze the effect of the ETC inhibitors on communication between centrosomes and mitochondria in M phase, we first treated cells with ROT, AA, Mdivi-1, or dynasore with simultaneous synchronization with nocodazole for 24 h. The presence or absence of nocodazole-treated cells served as the M phase- or interphase- mock control. In interphase, the Thr14 and Tyr15 residues of Cdk1 were phosphorylated by Wee1 and Myt1, which inhibited its activity [29]. Cell cycle arrest induced by ROT, AA, and Mdivi-1 was accompanied by a decrease in Cdk1 expression, but expression of Cyclin B was not affected by cell cycle arrest. It should also be noted that AA did not affect Cdk1 expression, indicating the existence of a discrepancy between AA and ROT/Mdivi-1. In contrast, Dynasore did substantially decrease Cyclin B expression at the G2/M phase and then proceed to next G1 cell cycle, and there was no apparent difference in Cdk1 levels after release (Figure 7A). It is thus clear that ROT, AA, and Mdivi-1, but not Dynasore suppressed Cyclin B nuclear translocation and delayed mitotic exit by blocking Cyclin B degradation (Figure 7A). 

Several mitotic kinases (Plks and Aurora A) are known to play roles in centriole duplication. A study also demonstrated that overexpression of Aurora A and Plk4 was associated with emergence of amplified centrosomes [6]. However, our results suggested that Aurora A, Plk1, and Plk4 protein levels were not affected under ROT or AA amplified centrosomes (Appendix A). Our previous studies have shown that AIBp also interacts with Plk1, which subsequently promotes Aurora-A–mediated Plk1 activation. Conversely, AIBp blocks the hNinein phosphorylation mediated by Aurora A and Plk1. Knockdown of AIBp expression caused down-regulation of Aurora A-Thr288, Plk1-Thr210 phosphorylation and mislocalization of ch-TOG to centrosomes, resulting in phenotypes with multiple spindles pole formation and chromosome misalignment. The interplay of hNinein, AIBp, Aurora A, and Plk1 possibly contributes to mitotic entry and bipolar spindles assembly during mitotic progression [30]. We next explored the molecular mechanisms and physiological functions of AIBp and Bora by looking at the downstream targets of Aurora A and Plk1, such as TACC3, ch-TOG, CEP192, and CEP215 in cell cycle arrest after ROT, AA, Mdivi-1, and Dynasore treatment. We found that ROT decreased the expression of AIBp and Bora in M phase, and downregulated TACC3, ch-TOG, CEP192, and CEP215 via phosphorylation of Plk1-T210 and Aurora A-T228 inhibition. Mdivi-1 treatment had the same effect as ROT on molecular mechanisms and physiological functions. Unexpectedly, AA also decreased the expression of Bora and AIBp, inhibited the phosphorylation of Aurora A-T228, and downregulated ACC3, ch-TOG, and CEP215 expression, but did not affect Plk1 phosphorylation and expression of CEP192, to cause multipolar spindles during M-phase (Figure 7B). These results suggest that ROT, AA, and Mdivi-1, but not Dynasore induce M-phase arrest to regulate spindles pole integrity and centrosome function.

## 4. Discussion

In this study, we provided evidence that Drp1 plays an essential role in the phosphorylation of Ser637, but not Ser616, to regulate mitophagy by inducing multipolar spindles in M phase. Markers for mitotic dysregulation, such as the aberrant protein expression of AIBp, Bora, Aurora A and Plk1, multipolar spindles and mitochondrial elongation, were observed in ROT, AA, and Mdivi-1 treated cells compared to Mock cells. Understanding this process will help delineate the relationship between maintenance of mitochondrial respiration and centrosomal abnormalities inducing spindles defects. A decrease in cell growth is associated with dysfunction in mitochondrial respiration. The blockade of electron transfer led to electron leaking. Of note, ETC complex IV blockade possibly caused reduced ATP synthesis [6,31]. In previous studies, Pallavi Srivastava and Panda demonstrated that the mitochondrial complex I inhibitor ROT inhibits microtubule assembly by attaching to the microtubules. The accumulation of BubR1 (an essential protein for spindles assembly checkpoint signaling) in the mitotic cells indicated that kinetochores were not properly attached to the microtubules [17,32]. In this study, we determined that mitochondrial dysfunction induces multipolar spindles formation by increased oxidative stress (Figure 1 and Figure 2). However, Dynasore had only a slight effect on mitotic defects and did not affect mitochondrial morphology (Figure 3). 

We found that two mitochondrial ETC inhibitors and Mdivi-1 significantly reduced Drp1 and phosphorylated Drp1-Ser616, and increased phosphorylated Drp1-Ser637, but Dynasore had no significant effect. In the past 10 years, there has been some progress in identifying and developing inhibitors of mitochondrial fission, including the molecules Mdivi-1, Dynasore, and mitochondrial division dynamin [24,32]. We chose to use Mdivi-1 and Dynasore. Nevertheless, similar to ROT inhibition, Mdivi-1, but not Dynasore caused multipolar spindles during M phase. Moreover, numerous studies have shown that the specificity of Dynasore as a selective inhibitor of dynamin GTPase activity to inhibit clathrin-mediated endocytosis has recently being disputed [33]. According to research Mdivi-1 may not be a specific Drp1 inhibitor but a selective Drp1 GTPase inhibitor, with the ability to reversibly inhibit complex I and modify mitochondrial ROS production [34]. Our results suggest that Mdivi-1 and Dynasore have discrepant biological activities. We noted that ETC inhibitors might have pleiotropic effects. It should be also noted that both rotenone and Mdivi-1 have been characterized with off-target effects (Rotenone inhibits microtubule polymerization; Mdivi-1 is a respiratory complex I inhibitor) [24,34]. In this study, we showed that mitochondrial dysfunction-induced aberrant centrosome amplification in HeLa cells is dependent on Drp1-Ser637 in M-phase. 

Previous reports showed that the activity of Drp1 is modulated by G2/M cell cycle arrest [35,36]. The effects of ROT on mitochondrial dysfunction in cancer have been reported [17]. Cell damage induced by AA results from the inhibition of electron transfer through the ETC [37], especially for the Mdivi-1-driven mitochondrial apoptotic pathway [38]. However, to our knowledge, there is no previous study examining the relationship between mitochondrial fission and cell cycle after ROT, AA, and Mdivi-1 administration in HeLa cells. To verify whether mitochondrial ETC inhibitor-induced G2/M cell cycle arrest is mediated by Drp1 in cervical cancer cells, we investigated if downregulation of Drp1 could reduce Mdivi-1 and Dynasore-induced cell G2/M cycle arrest. Mdivi-1 and Dynasore aggravated the mitochondrial fragmentation causing multipolar spindles formation, indicating that induction of multipolar spindles in mitotic cells is associated with mitochondrial ETC blockade (Figure 4F). These results further validate the vital role of the mitochondrial fission/fusion dynamic in mitochondrial ETC blockade-mediated G2/M cell cycle arrest in HeLa cells.

In the G2/M phase, mitochondria undergo fragmentation by Cdk1/Cyclin B which directly phosphorylates at Drp1-Ser616 [39]. Since fragmentation is reported to induce mitochondrial depolarization, the observed depolarization might be due to fragmentation rather than a decrease in the supply of electrons to the ETC [26]. Mitotic fragmentation of mitochondria is mediated by Cdk1-dependent phosphorylation and Drp1 activation, involved in mitochondrial fission [40]. Abnormal mitochondrial fusion induces fragmentation of mitochondrial tubules [41,42]. However, mitochondrial fragmentation occurs in more than one way. (1) Excessive ROS production and reduced ATP production shatters mitochondrial tubules; and (2) phosphorylated Drp1 at Ser616 and Ser637 sites alters GTPase activity, causing defective mitochondrial fission. In this study, we showed that inhibiting the Aurora A-Plk1 cascade induced mitochondrial dysfunction and aberrant centrosome amplification. This suggests the involvement of these kinases in mitochondrial dysfunction-induced defective/deregulated mitosis. In general, M-phase exit in the cell cycle is controlled via three different mitotic kinases, Cdk1/Cyclin B, Aurora A, and Plk1, to regulate mitochondria. Cdk1/Cyclin B directly phosphorylates Drp1-Ser616; Aurora A recruits Drp1 via PALA/RABP1; and Plk1 phosphorylates Mito/CenpF [21]. Aurora A directly regulates mitochondria fragmentation and fusion through its protein level imported into mitochondria during interphase, whereas it promotes mitochondrial fission in mitosis [23,43]. Furthermore, the mitochondrial fission mediator Drp1 appears to antagonize Cdk1/Cyclin B by promoting slippage while the elevated ROS level correlates with higher Cdk1/Cyclin B activity favoring a sustained mitotic arrest. Thus, the fate of a mitotically arrested cell is determined by the complex interactions between various regulatory modules [30].

According to the aforementioned results, we propose a model of possible paths for communication between centrosomal and mitochondrial organelles during bipolar spindles formation (Figure 8 and Appendix A). By identifying mitophagy as a mechanistic link between mitochondria and centrosome organelles, our findings may offer a new conceptual view for how mitochondrial proteins act in multipolar spindles formation. The coupling of mitochondrial ETC proteins and centrosomal kinases/proteins may be important for centrosome function and bipolar spindles assembly. After nocodazol treatment, during the M-phase arrest, four inhibitors (two ETC inhibitors, ROT and AA, and two Drp1 inhibitors, Mdivi-1 and Dynasore) were added to induce mitochondrial dynamic imbalance. The results showed that mito-Drp1-Ser637 rather than Drp1-Ser616 via the PINK1/Parkin pathway (Figure 6) affected M-phase kinases (Cdk1, Plk1, and Aurora A and their associated centrosomal proteins Cyclin B, Bora, AIBp, CEP192/CEP215, and TACC3/chTOG (Figure 7) [30,44] to cause multipolar spindles formation. It should be noted that in general, phosphorylation at Ser637 inhibits Drp1 activity while phosphorylation at Ser616 activates Drp1; some specific upstream kinases also influence Drp1 function [41,42,45]. We also noted a recent study addressing the status of Drp1 phosphorylation at Ser637 (Drp1-Ser637) for subcellular localization to both the cytosol and mitochondria, supporting the notion that Drp1-Ser637 can shuttle between the cytosol and mitochondria and indicating that Drp1-Ser637 does not play a major role in controlling mitochondrial fission [46]. The data are somehow at odds with the view that Drp1 phosphorylated at amino acid Ser637 by PKA is retained in the cytosol [30,46,47]. Nevertheless, in comparison to these studies, our present study of FSK (PKA activator) or H89 (PKA inhibitor) showed phosphorylation of Drp1-Ser637 rather than Drp1-Ser616 in a PKA-dependent performance at mitosis, suggesting that mito-Drp1-Ser637 indeed plays a major role in multipolar spindles formation during insults by inhibitors to cause mitotic arrest (Figure 5). Moreover, some studies have shown that cells were protected from diverse insults through remodeling by PKA/AKAP1 [48] or PKA/GSKIP [49] or even PKA/AKAP220 [50]. In support of our findings using ETC inhibitors and Mdivi-1 insults in this study, it is more likely that mitochondrial dynamics are compromised by both Drp1/PKA/AKAPs and Cdk1/Cyclin B signaling axes (left side in Appendix A). It should be also noted that the resistance to mitotic cell death from multipolar spindles formation acquired upon Drp1-Ser637 activation supports the speculated ability of fragmented mitotic mitochondria to escape cell death (apoptosis) via mitophagy. However, deregulation of the Plk1/Cdk1/Cyclin B axis to cause mitochondrial hyperfusion may exert opposite effects, leading to errors in mitosis entry and mitotic stress. It is believed that Drp1-Ser616 fission activity regulated by Cdk1/Cyclin B is a prerequisite for normal mitochondrial partition and distribution into two daughter cells at mitosis ([39], Figure 8). Nevertheless, when cells are insulted by ROT, AA, and Mdivi-1 inhibitors, the phosphorylation status of mito-Drp1-Ser637 by PKA rather than mito-Drp1-Ser616 by Cdk1/Cyclin B in mitochondrial fission modulates mitophagy to induce multipolar spindles assembly at mitotic arrest (Figure 8). We also note that PKA may influence mitochondrial dynamics in multiple ways and alterations in mitochondrial morphology should not be regarded simply as a functional consequence of the mito-Drp1–Ser637 phosphorylation. However, our data show here for the first time that mito-Drp1-Ser637 instead of Drp1-Ser616 at M-phase plays a major role dictating mitochondrial fission, inducing mitotic kinases via the PINK1/Parkin pathway for ongoing mitophagy. Since decreased membrane potential, reduced ATP production, and increased ROS are the three common hallmarks triggering apoptosis, ongoing mitophagy during mitotic arrest (by mito-Drp1-Ser637) may simply represent leaky degradation from incompletely blocked autophagy, which is functionally relevant during prolonged mitotic arrest but likely negligible during normal mitotic progression (Drp1-Ser616). Eventually, this mechanism may also participate in pushing cells with mitotic abnormalities toward cell aneuploidy or cell death (Figure 8 and right side in Appendix A) [11,35,51,52,53,54]. We showed that ETC and Drp1 inhibitors affect the dynamic interplay between mito-Drp1-Ser637 and mito-Drp1-Ser616 (Ser637/Ser616 ratio) via the PINK1/Parkin pathway to drive mitophagy. Finally, disrupting M-phase kinases/proteins (Cdk1, Plk1, and Aurora A and their associated centrosomal proteins Cyclin B, Bora, AIBp, CEP192/CEP215, and TACC3/chTOG) induced multipolar spindles formation (labeled with a red box in Appendix A). How mitochondrial Drp1-Ser637 directly induced by PKA/AKAP specifically confers with centrosomal-related kinases/proteins directly insulted by inhibitors (red box and upside-down, in Appendix A) warrants further detailed molecular studies. 

## 5. Conclusions

Our data show that crosstalk between mitochondrial ETC proteins and centrosomal kinases/proteins plays an important role in centrosome function and bipolar spindles assembly. At normal cell cycle stages, Drp1-Ser616 drives faithful mitochondrial partition, but after insult mito-Drp1-Ser637 drives mitophagy via activated PKA during mitotic arrest. The net results are that mito-Drp1-Ser637 induces mitochondrial fission to drive mitophagy and induce multipolar spindles formation (Figure 8). Monitoring evoked mito-Drp1-Ser637 rather than Drp1-Ser616 in mitochondria along with centrosome amplification could represent a potential pharmacodynamic biomarker for future therapeutics aimed at restoring normal mitochondrial partition or mitophagy during mitosis.

## Figures and Tables

**Figure 1 biomolecules-11-00424-f001:**
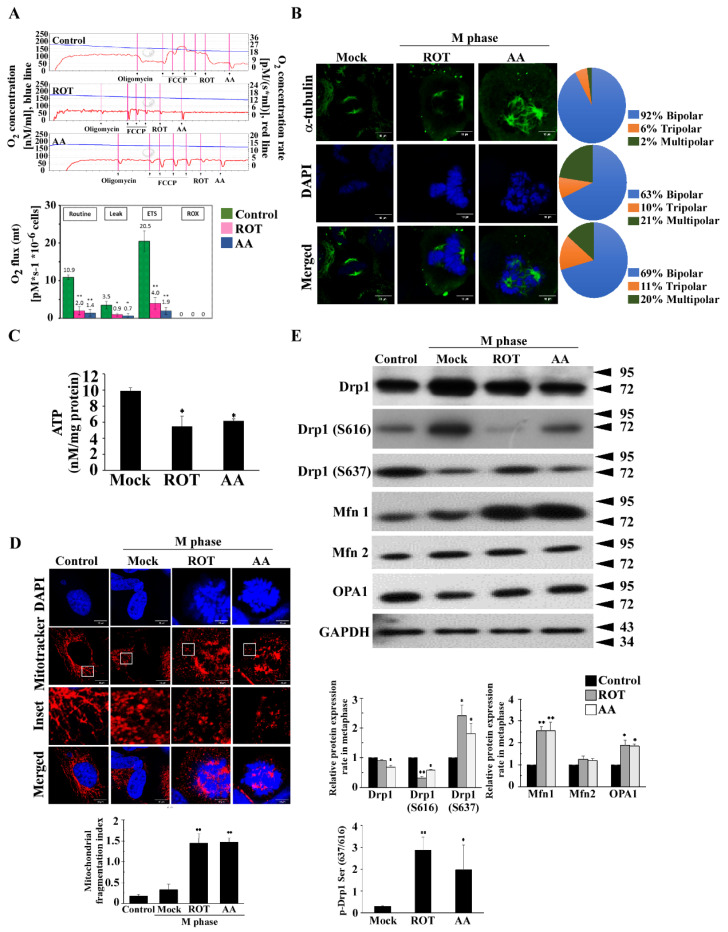
Inhibition of mitochondrial ETC complexes I and III induces multiple spindles formation. (**A**) HeLa cells lost almost all respiratory capacity after ROT or AA treatment. The OCR was determined in the presence of ROT (200 nM), AA (10 μM) and control at 24 h using an oxygen electrode (Ouroboros O2k). The PCP protocol (phosphorylation control protocol) was applied to evaluate OCR, routine, leak, ETS (maximum electron transfer system) and ROX (residual oxygen consumption) values in ROT (pink) or AA (blue) treated HeLa cells compared to control (green) HeLa cells using O_2_k. (**B**) HeLa cells were synchronized at G2/M by nocodazole. More than 300 cells were calculated. Quantitation pie charts indicate the percentage of bipolar, tripolar, and multipolar mitotic cells. (**C**) Normalized ATP measurements in HeLa cells. (**D**) Mitochondrial morphological features in the presence of ROT or AA were evaluated by the changes in mitochondrial dynamics. The mitochondria were stained with MitoTracker Red to evaluate mitochondrial morphology. Blue DAPI represents chromosome. (**E**) HeLa cells were analyzed for mitochondrial dynamics-associated proteins by treating with ROT and AA for 24 h synchronized with nocodazole and then released into the M phase. GAPDH was used as an internal control. The bar graph represents the mean of triplicates ± SD. * *p* < 0.05, ** *p* < 0.01 compared with the Mock group.

**Figure 2 biomolecules-11-00424-f002:**
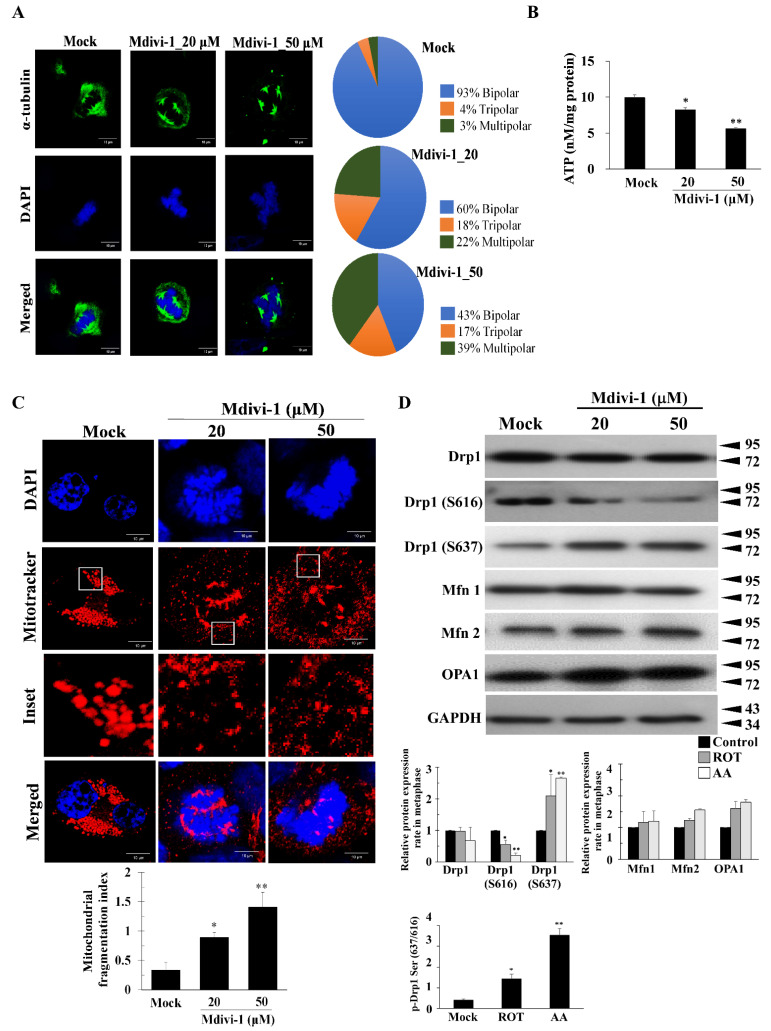
Mdivi-1 causes the formation of multiple spindles poles and uneven segregation. (**A**) HeLa cells were synchronized at G2/M by nocodazole, followed by the incubation for 24 h in drug-free medium in the absence (Mock) or presence of Mdivi-1. More than 300 cells were calculated. Quantitation pie charts indicate the percentage of bipolar, tripolar, and multipolar mitotic cells. (**B**) Normalized ATP measurements in HeLa cells. (**C**) HeLa cells were synchronized by either a double nocodazole block, release into fresh medium at the indicated time points, and then analysis by immunoblotting with antibodies against the indicated proteins. (**D**) Mitochondrial morphological features in the presence of Mdivi-1 were evaluated by the changes in mitochondrial dynamics. GAPDH was used as an internal control. The bar graph represents the mean of triplicates ± SD. * *p* < 0.05, ** *p* < 0.01 compared with the Mock group.

**Figure 3 biomolecules-11-00424-f003:**
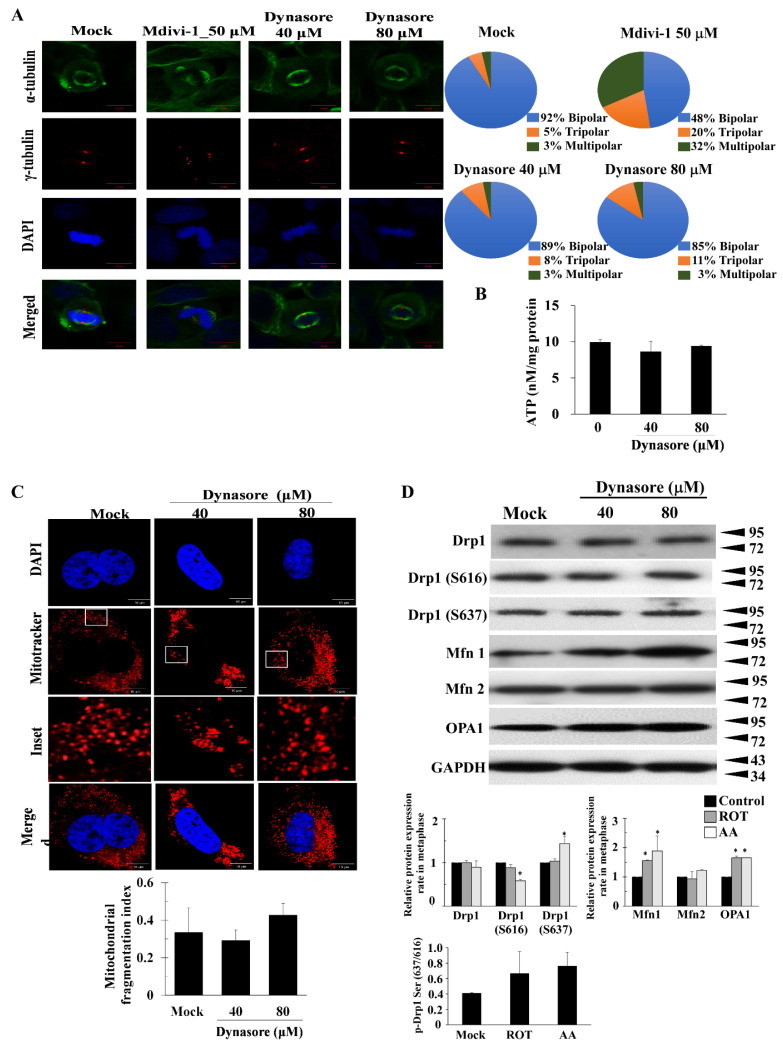
Dynasore did not significantly affect the mitotic defects. (**A**) HeLa cells were synchronized at G2/M by nocodazole, followed by incubation for 24 h in drug-free medium in the absence (control) or presence of Dynasore. More than 300 cells were calculated. Quantitation pie charts indicate the percentage of bipolar, tripolar, and multipolar mitotic cells. (**B**) Normalized ATP measurements in HeLa cells. (**C**) Mitochondrial morphological features in the presence of Dynasore were evaluated by the changes in mitochondrial dynamics. (**D**) HeLa cells were synchronized by double nocodazole block, released into fresh medium at the indicated time points, and then analyzed by immunoblotting with antibodies against the indicated proteins. GAPDH was used as an internal control. The bar graph represents the mean of triplicates ± SD. * *p* < 0.05 compared with the Mock group.

**Figure 4 biomolecules-11-00424-f004:**
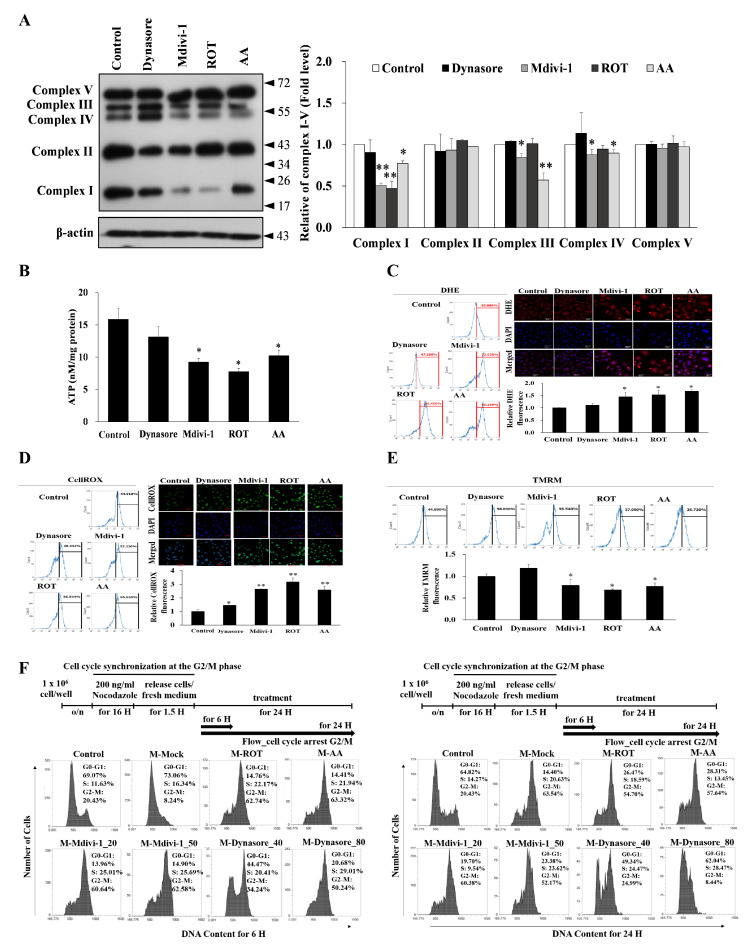
Increase in multipolar spindles is associated with mitochondrial ETC blockade and cell-cycle progression in HeLa cells. (**A**) Immunoblotting analysis of OXPHOS mitochondrial complexes in ROT (200 nM), AA (10 μM), Mdivi-1 (20 μM), and Dynasore (40 μM) treated HeLa cells. (**B**) Normalized ATP measurements in HeLa cells. (**C**–**E**) The mitochondrial ROS, cytosolic ROS and mitochondrial membrane potential (ΔΨm) levels were measured using flow cytometry and fluorescence microscopy with DHE Deep Red, CellROX Deep green, and TMRM Deep Red staining, respectively, and indicated by the mean fluorescent intensity (MFI), or percentage of the cells. (**F**) Mitochondrial ETC delays cell-cycle G2–M to G1 phase transition. FACS analyses were carried out using cells released from nocodazole block in the presence of ROT, AA, Mdivi-1, and Dynasore for 16 and 24 h. Cell cycle distribution was determined by flow cytometric analysis of propidium iodide-stained cells. Cell number (%) in each cell-cycle phase is indicated in the graph. The bar graph represents the mean of triplicates ± SD. * *p* < 0.05, ** *p* < 0.01 compared with the control group.

**Figure 5 biomolecules-11-00424-f005:**
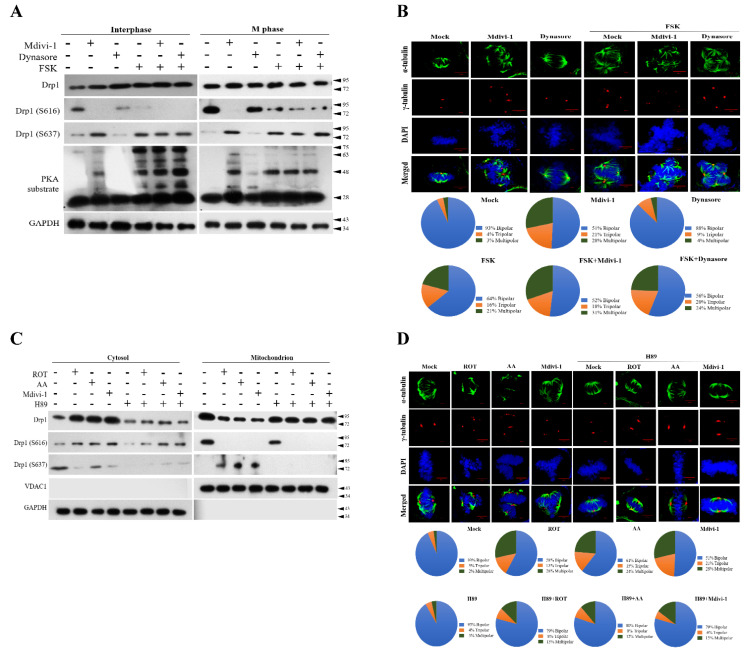
Blockade of mitochondrial ETC and morphological features by ROT, AA, Mdivi-1, and Dynasore along with the presence of FSK or H89 results in phosphorylation of Drp1-Ser637 and Drp1-Ser616 in PKA performance at mitosis. (**A**) FSK-enhanced Dynasore stimulated Drp1-dependent mitochondrial fission. (**B**) FSK-enhanced Dynasore induced multiple spindles formation. (**C**) H-89 suppressed the mito-Drp1-S637 phosphorylation induced by ROT, AA, and Mdivi-1. (**D**) H-89 inhibited multiple spindles formation induced by ROT, AA, and Mdivi-1. Cells were pretreated with the PKA inhibitor H89 (50 μM) or PKA activator FSK (25 μM) for 2 h, followed by the treatment with ROT, AA Mdivi-1, and Dynasore for 24 h; whole-cell lysates were then prepared and subjected to Western blot. Quantitation pie charts indicate the percentage of bipolar, tripolar, and multipolar mitotic cells. VDAC1 and GAPDH were used as mitochondria and cytosol markers, respectively. GAPDH was checked for subcellular purity.

**Figure 6 biomolecules-11-00424-f006:**
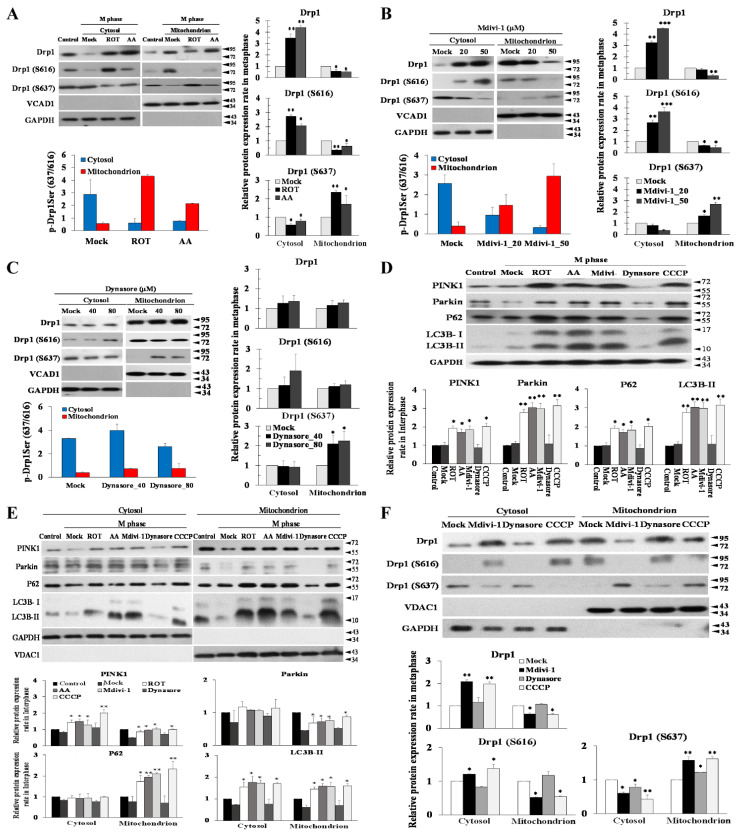
Blockade of mitochondrial ETC results in phosphorylation of mito-Drp1 Ser-637, which is essential for the PINK1/Parkin pathway to regulate mitophagy. (**A**) ROT or AA decreased Drp1-dependent mitochondrial fission. (**B**) Mdivi-1 decreased Drp1-dependent mitochondrial fission. (**C**) Dynasore is not a specific Drp1 inhibitor. Levels of total Drp1, Drp1-Ser616, and Drp1-Ser637 were checked in cytoplasmic and mitochondrial fractions in M phase with ROT, AA, Mdivi-1, and Dynasore for 24 h. (**D**) ROT, AA, and Mdivi-1 treatments can upregulate expression of Parkin and promote mitophagy in M-phase arrest. (**E**) Levels of PINK1/Parkin, P62, and LC3B were checked in cytoplasmic and mitochondrial fractions in M phase with ROT, AA, Mdivi-1, and Dynasore. (**F**) CCCP decreased Drp1-dependent mitochondrial fission. VDAC1 was assessed as an internal loading control. GAPDH was checked for subcellular purity. The bar graph represents the mean of triplicates ± SD. * *p* < 0.05, ** *p* < 0.01, *** *p* < 0.001 compared with the control group and Mock group.

**Figure 7 biomolecules-11-00424-f007:**
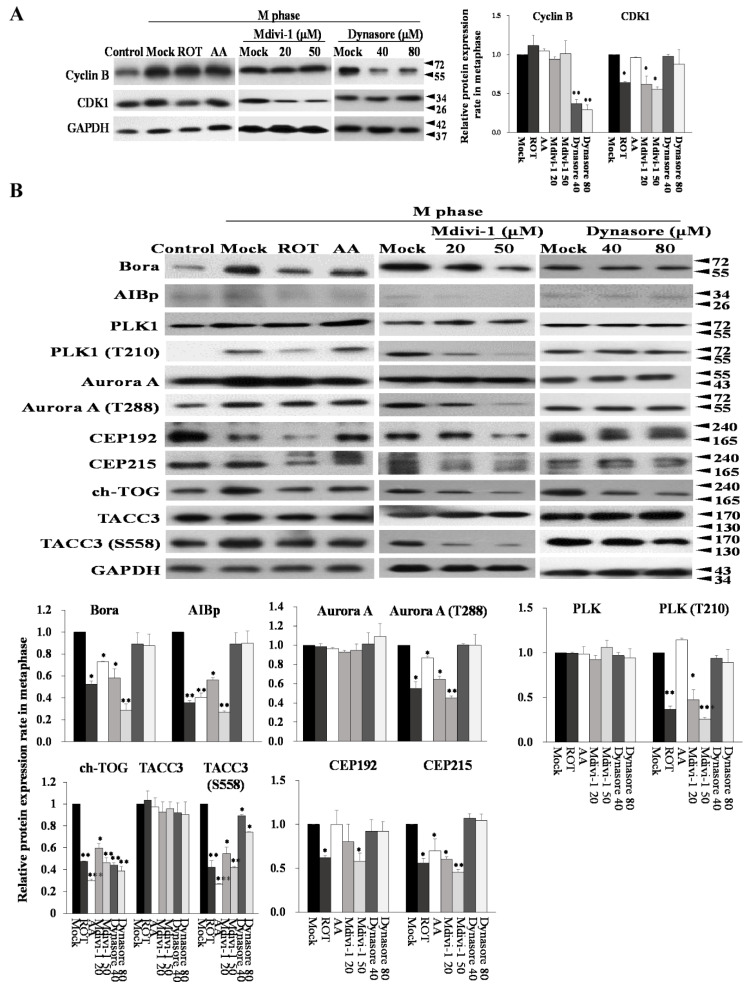
Blockade of mitochondrial ETC results in aberrant performance of the Aurora A/Plk1 cascade in M phase. (**A**) HeLa cells were either synchronized by a nocodazole block or were treated with ROT, AA, Mdivi-1, and Dynasore for 24 h, followed by Western blot analysis of key cell cycle regulators. (**B**) HeLa cells were synchronized in G2/M by 16 h nocodazole followed by release. Cells were treated as indicated. The Aurora A/Plk1 cascade was immunoprecipitated and samples were probed with the indicated antibodies. GAPDH was used as an internal control. The bar graph represents the mean of triplicates ± SD. * *p* < 0.05, ** *p* < 0.01, *** *p* < 0.001 compared with the control group. Note that all Mock lanes also as normalized M phase.

**Figure 8 biomolecules-11-00424-f008:**
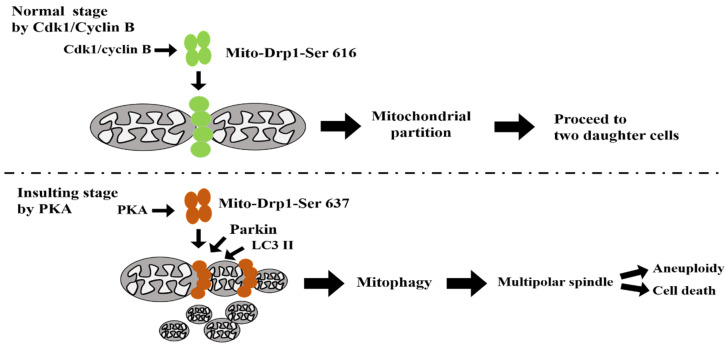
Model of cross-talk between mitochondrial ETC proteins and centrosomal kinases/proteins for centrosome function and bipolar spindles assembly. After nocodazole treatment, at M phase four inhibitors (two ETC inhibitors, ROT and AA; two Drp1 inhibitors; Mdivi-1 and Dynasore) were added to induce mitochondrial dynamic imbalance. AKAP1 is believed to be a scaffold protein that recruits PKA to modulate mitochondrial function at Drp1-Ser637 (also see Appendix A). In summary, upon normal conditions, during mitosis mitochondria are fragmented to facilitate segregation of this organelle to daughter cells. Mitochondrial fission is mediated by Cdk1-dependent phosphorylation and activation of Drp1-Ser616 [39]. Proper partitioning of mitochondria is required for inheritance during cell division, and then proceeds to the next cell cycle. However, upon inhibitor insulting conditions, mito-Drp1-Ser637 drives mitophagy via activated PKA during mitotic arrest. Dynamic Drp1-Ser637/Ser616 ratio affects the PINK1/Parkin pathway and promotes more efficient mitophagy while inducing multipolar spindles formation simultaneously. It appears that mito-Drp1-Ser637 induces mitochondrial fission to drive mitophagy and induces multipolar spindles formation to cause cell aneuploidy or cell death.

## Data Availability

Data is contained within the article or Appendix A.

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
