# Peer review of "The Phosphorylation Status of Drp1-Ser637 by PKA in Mitochondrial Fission Modulates Mitophagy via PINK1/Parkin to Exert Multipolar Spindles Assembly during Mitosis"

_biomolecules, 2021, doi:10.3390/biom11030424_

Round 1
Reviewer 1 Report
In the manuscript entitled: The phosphorylation status of Drp1-Ser637 by PKA in mitochondrial fission modulates mitophagy via PINK1/Parkin to exert multipolar spindles assembly during mitosis Huey-Jiun Ko and colleagues investigate the molecular mechanism of mitophagy, a phenomenon similar to autophagy, by which the cells prevent the accumulation of dysfunctional mitochondria. Starting by the fact that the loss of protein Drp1, essential in the mechanism of fission mitochondrial, causes mitochondrial hyperfusion and induces aneuploidy the authors investigate an interesting and timely theme on the crosstalk between mitochondria and centrosome.
The work is of interest and original here below some concerns on the methodology and interpretations of results
Major points
- Please explains the rational of using rotenone and antimycin A, AA when describing them introducing the results. Perhaps many experiments showed in Figure 1 and 2 may be placed in supplementary leaving space for more original results showed in the final part of the study (Figure 6,7)
- In the first part of the manuscript, the authors often use the terms centrosome amplification but it is not clear which centrosome markers are indeed increased.
- The percentage of mitotic defects should be related to the number of cells indeed measured or counted using a more rigorous statistical approach
- In the (Figures 1 and 2) 1G 2F etc… the images representing treated cells vs control and Mock cannot be compared because they are of cells in interphase (controls) vs cells (treated) undergoing in mitosis. Improving this data is very important for not reduces the impact and rationale of article.
- In Figure 1 and Figure 2 the loading control for western blot analysis is not convincing in relation to the number of analyzed markers. Next the phosphorylated form of Drp1 (control vs Mock) are to different. Please replace it or explains what is really happening.
- The western blot images provided in Figure 2G in comparison to those showed in Figure 1 do not contain the control. Please Uniform all western blot experiments.
- While the increased phosphorylation of Drp1 is interesting, however, remains unclear how it changes with centrosome localization and cell cycle progression (interphase vs mitosis).
- According to the comments provided above, Please make attention to the following sentence (we observed an increase in mitochondrial fragmentation in Mdivi-1 treated cells in nocodazole-induced mitotic arrest (Figure 2F).
- Please revise Figure 3A. The sentence on “Dynasore slightly promoted mitochondrial superoxide formation in M phase should be consistently proved by western blot on different cell cycle phases.
- The data presented in Figure 5 are of interest but should be more clearly showed and interpreted. For example the western for interphase is not clear. Next, the experiments (western blot) should be performed comparing the effect (FSK,PKA activator) and H89 (PKA inhibitor).
- Next it is not clear what experimental support is provided to state that Drp1-Ser616 fission activity regulated by Cdk1/Cyclin B is a prerequisite for normal equal mitochondria partition and faithful distribution into two dividing cells at M-phase. Although a reference is provided the effect of Cdk1/Cyclin B, no study is done on (CDK1 inhibitors or its genetic silencing). The paper in fact suffers of any study of specific gene silencing or site specific mutagenesis on the involved Ser aa.
- With the available data, the authors fail to show if a direct relationship between mitochondria and centrosome exists. Through the manuscript the concept of centrosome amplification introduced in Figure 1 but it is contradicting when authors measures the centrosome markers in Figures 6 and 7. This contradiction emerges also by the The fact to have two models in Figure 8. Please simplify it.
Minor Points
- Please polished some language mistakes throughout the manuscript.
- Please provide higher quality western-blot, loading bands and IF images for critical data.
- Often some sentences in the manuscript are too long and not clearly structured.
- The discussion is too long.
Author Response
Responses to reviewer comments:
In the manuscript entitled: The phosphorylation status of Drp1-Ser637 by PKA in mitochondrial fission modulates mitophagy via PINK1/Parkin to exert multipolar spindles assembly during mitosis Huey-Jiun Ko and colleagues investigate the molecular mechanism of mitophagy, a phenomenon similar to autophagy, by which the cells prevent the accumulation of dysfunctional mitochondria. Starting by the fact that the loss of protein Drp1, essential in the mechanism of fission mitochondrial, causes mitochondrial hyperfusion and induces aneuploidy the authors investigate an interesting and timely theme on the crosstalk between mitochondria and centrosome.
The work is of interest and original here below some concerns on the methodology and interpretations of results
Response to reviewer 1:
To answer point-by-point the details of the revisions to reviewer 1:
Major points
Q1. Please explains the rational of using rotenone and antimycin A, AA when describing them introducing the results. Perhaps many experiments showed in Figure 1 and 2 may be placed in supplementary leaving space for more original results showed in the final part of the study (Figure 6,7).
Ans: Rotenone (complex I inhibitor) and antimycin A (AA, complex III inhibitor) are common inhibitors to study mitochondrial electron transport chain (ETC). Please also see rotenone and AA as inhibitors on references 6, 11 and 17. Besides, we agree to move some experiments in figure1 and 2 into supplementary files.
- In the first part of the manuscript, the authors often use the terms centrosome amplification but it is not clear which centrosome markers are indeed increased.
Ans: Our data revealed mislocation of centrosome markers (Cdk1, Plk1 and Aurora A and their associated centrosomal proteins Cyclin B, Bora, AIBp, CEP192/CEP215 and TACC3/chTOG) instead of increase in their expression levels. Only PKA activity and p-Drp1-637 were significantly increased in our study
- The percentage of mitotic defects should be related to the number of cells indeed measured or counted using a more rigorous statistical approach
Ans: We also used Image-J for counting number of cells to confirm, we think it was suitable for measuring.
- In the (Figures 1 and 2) 1G 2F etc… the images representing treated cells vs control and Mock cannot be compared because they are of cells in interphase (controls) vs cells (treated) undergoing in mitosis. Improving this data is very important for not reduces the impact and rationale of article.
Ans: Interphase (we named it as control group) was used for negative control. True control is Mock group in M phase. All data were compared to Mock (Mitosis) not to interphase (Control group).
- In Figure 1 and Figure 2 the loading control for western blot analysis is not convincing in relation to the number of analyzed markers. Next the phosphorylated form of Drp1 (control vs Mock) are to different. Please replace it or explains what is really happening.
Ans: Please see our supplemental Figure 1D, indicating the experiments can be manipulated on M phase, which can tell the differences between Interphase and M phase. To avoid the confuse with Mock, we eliminate the interphase data in figure 2. Of note, Interphase (we named it as control group) was used for negative control. True control is Mock group in M phase. All data were compared to Mock (Mitosis) not to interphase (Control group). So phosphorylated form of Drp1(control vs Mock) are different in nature.
Q6. The western blot images provided in Figure 2G in comparison to those showed in Figure 1 do not contain the control. Please Uniform all western blot experiments.
Ans: Interphase (we named it as control group) was used for negative control. True control is Mock group in M phase. All data were compared to Mock (Mitosis) not to interphase (Control group). So we cancelled the control group in former Figure 2G (has been changed to Figure 2D). Please also see above answer 5.
Q7. While the increased phosphorylation of Drp1 is interesting, however, remains unclear how it changes with centrosome localization and cell cycle progression (interphase vs mitosis).
Ans: Your question is well taken. Indeed, the data are somehow at odds with the view that Drp1 phosphorylated at amino acid Ser637 by PKA is retained in the cytosol [30,47,48]. We also noted a recent study addressing the status of Drp1 phosphorylation at Ser637 (Drp1-Ser637) for subcellular localization to both the cytosol and mitochondria, supporting the notion that Drp1-Ser637 can shuttle between the cytosol and mitochondria and indicating that Drp1-Ser637 does not play a major role in controlling mitochondrial fission [47]. Please also see above answer 5 to tell the difference between interphase and M phase and we emphasize M phase in this study. In the future, we will keep an eye on the mitophagy function of Drp1-637 in interphase and directly affect to exert multiple spindle formation in M phase (also see the section of Discussion lines 683-686.)
Q8. According to the comments provided above, Please make attention to the following sentence (we observed an increase in mitochondrial fragmentation in Mdivi-1 treated cells in nocodazole-induced mitotic arrest (Figure 2F).
Ans: We agree. Our data showed that Mdivi-1(former Figure 2F, now changed to Figure 2C) is similar with rotenone (former Figure 1G, now changed to Figure 1D) in nocodazole-induced mitotic arrest. Please also see reference 35 “The Putative Drp1 Inhibitor mdivi-1 Is a Reversible Mitochondrial Complex I Inhibitor that Modulates Reactive Oxygen Species. Dev Cell 2017” is consistent to our observations in this study for additional function of Mdivi-1 similar to Rotenone function.
- Please revise Figure 3A. The sentence on “Dynasore slightly promoted mitochondrial superoxide formation in M phase should be consistently proved by western blot on different cell cycle phases.
Ans: We changed it as “Dynasore slightly promoted multipolar spindles formation in M phase” in lines 351-352.
Q10. The data presented in Figure 5 are of interest but should be more clearly showed and interpreted. For example the western for interphase is not clear. Next, the experiments (western blot) should be performed comparing the effect (FSK,PKA activator) and H89 (PKA inhibitor).
Ans: Your question is well taken. We think both phases (interphase and mitosis) have the same phenomenon. In the future, we will keep an eye on the mitophagy function of Drp1-637, but not Drp1-616 in interphase.
Q11. Next it is not clear what experimental support is provided to state that Drp1-Ser616 fission activity regulated by Cdk1/Cyclin B is a prerequisite for normal equal mitochondria partition and faithful distribution into two dividing cells at M-phase. Although a reference is provided the effect of Cdk1/Cyclin B, no study is done on (CDK1 inhibitors or its genetic silencing). The paper in fact suffers of any study of specific gene silencing or site specific mutagenesis on the involved Ser aa.
Ans: Your question is well taken. In the future, we will perform the further experiments such as Drp1-637 mutant A-form, mimic D-form, inhibitors and siRNA, as well as Drp1-616 to further prove our findings.
Q12. With the available data, the authors fail to show if a direct relationship between mitochondria and centrosome exists. Through the manuscript the concept of centrosome amplification introduced in Figure 1 but it is contradicting when authors measures the centrosome markers in Figures 6 and 7. This contradiction emerges also by the The fact to have two models in Figure 8. Please simplify it.
Ans: Your question is well taken. It is well-known that Aurora A and Plk1 locate and phosphorylate LARA and Miro in mitochondria, respectively [19-21]. Therefore, we speculated that PLK1 and Aurora A phosphorylated Drp1 with direct relationship in our supplemental data model. We also agree your comment to move Figure 8A to supplementary Figure 8. We also simplified the discussion and maintained Figure 8B model as Figure 8.
Minor Points
Q13. Please polished some language mistakes throughout the manuscript.
Ans: We already polished and corrected it.
Q14. Please provide higher quality western-blot, loading bands and IF images for critical data.
Ans: We already provided high quality data and original files to editors.
Q15. Often some sentences in the manuscript are too long and not clearly structured. The discussion is too long.
Ans: Your question is well taken. We already polished and corrected it. We also simplified the discussion

Reviewer 2 Report
In this manuscript, authors investigated the phosphorylation status of mitochondrial fission protein Drp1 by PKA can modulate mitophagy via PINK1 pathway during mitosis. Authors found that the phosphorylation of Drp1-Ser637 by PKA is more vital effect rather than Drp1-Ser616 during formation of multipolar spindles in M-phase. This study is based on the fact that PKA phosphorylates Drp1 at Ser637 which has been reported previously. Inhibition of ETC causes Drp1-Ser637 inactivity following mitochondrial hyperfusion and fission inhibition resulting chromosome instability by multipolar spindle formation at G2/M phase. This study is unique in its core provides insights of mitochondrion’s role in cell cycle exit and which can be further implemented in cell differentiation.
Comments to authors:
- In line 320, authors used Fig. 3A instead of Figure 3A. please check it.
- In Figure 1H, ambiguous presentation of quantification data and missing Mfn1, 2 and OPA data.
- In Figure 2G, authors should address that why pDrp1-S637 increased as Mdivi-1 Drp1 inhibitor?
- Dynasore vs Mdivi-1 (I am wondering Dynasore has slight negative effect on mitochondrial fission and morphology as it is also Drp1 inhibitor)
- In Figure 6F, the expression of Drp1 is detected in inhibitor treated HeLa cells. Authors should be addressed.
- Discussion is too long and wrote discussion like results with figure number. Authors should consider this concern.
- Please check the reference style, No DOI.
Author Response
Responses to reviewer comments:
Response to reviewer 2:
In this manuscript, authors investigated the phosphorylation status of mitochondrial fission protein Drp1 by PKA can modulate mitophagy via PINK1 pathway during mitosis. Authors found that the phosphorylation of Drp1-Ser637 by PKA is more vital effect rather than Drp1-Ser616 during formation of multipolar spindles in M-phase. This study is based on the fact that PKA phosphorylates Drp1 at Ser637 which has been reported previously. Inhibition of ETC causes Drp1-Ser637 inactivity following mitochondrial hyperfusion and fission inhibition resulting chromosome instability by multipolar spindle formation at G2/M phase. This study is unique in its core provides insights of mitochondrion’s role in cell cycle exit and which can be further implemented in cell differentiation.
To answer point-by-point the details of the revisions to reviewer 2:
Q1. In line 320, authors used Fig. 3A instead of Figure 3A. please check it.
Ans: We changed Fig.3A to Figure 3A
Q2. In Figure 1H, ambiguous presentation of quantification data and missing Mfn1, 2 and OPA data.
Ans: We added the quantification of Mfn1/2 and OPA data in former Figure 1H, now has been moved to Figure 1E.
Q3. In Figure 2G, authors should address that why pDrp1-S637 increased as Mdivi-1 Drp1 inhibitor?
Ans: Sorry for our mistake, we misspelling Figure 2G (now has been changed to Figure 2D) in line 337. So we changed it to Figure 2C and also already addressed the explanations in original article in lines 337-342. Please also refer to Figure 5A. in M phase, lane 2, which indicating Mdivi-1 alone can truly increase PKA activity to pDrp1-S637
Q4. Dynasore vs Mdivi-1 (I am wondering Dynasore has slight negative effect on mitochondrial fission and morphology as it is also Drp1 inhibitor)
Ans: Although the sentence “Dynasore did not affect mitochondrial morphology” in our manuscript, but please see reference 34. Dynasore - not just a dynamin inhibitor. Dynasore is dynamin I/II/III inhibitor, but Mdivi-1 is dynamin IV (Drp1) inhibitor. So Mdivi-1 and Dynasore have discrepant biological activities.
Q5. In Figure 6F, the expression of Drp1 is detected in inhibitor treated HeLa cells. Authors should be addressed.
Ans: In Figure 6F, the results of this study were consistent with previous studies showing that 10 μM CCCP decreased mito-Drp1-Ser616 and mito-Drp1 and increased mito-Drp1-Ser637 levels at mitosis (Figure 6F), further confirming the involvement of PINK1/Parkin in mito-Drp1-Ser637 induced mitophagy. We defined the status of Drp1 phosphorylation at Ser637 in mitochondria by protein kinase A (PKA) (herein designated mito-Drp1-Ser637) in the section of Introduction in lines 103-104.
Q6. Discussion is too long and wrote discussion like results with figure number. Authors should consider this concern.
Ans: We agree to shorten the partial section of discussion.
Q7. Please check the reference style, No DOI.
Ans: We already checked it!

Round 2
Reviewer 1 Report
The authors have promptly replied to the raised questions improving the quality of the manuscript
Author Response
Response to reviewer 1:
The authors have promptly replied to the raised questions improving the quality of the manuscript.
Ans: Thank you for accepting our article.
This manuscript is a resubmission of an earlier submission. The following is a list of the peer review reports and author responses from that submission.